# Fabrication and Evaluation of W/O Emulsion Loaded with *Linum usitatissimum* Seeds Extract for Anti-Leishmaniasis Efficacy

**DOI:** 10.3390/antibiotics11040432

**Published:** 2022-03-23

**Authors:** Barkat Ali Khan, Sumera Faiz, Muhammad Khalid Khan, Farid Menaa, Neli-Kinga Olah, Yosif Almoshari, Jawaher Abdullah Alamoudi, Saud Almawash

**Affiliations:** 1Drug Delivery and Cosmetics Lab (DDCL), Faculty of Pharmacy, Gomal University, Dera Ismail Khan 29050, Pakistan; sumerafaiz77@gmail.com (S.F.); khalid.gomalian@gmail.com (M.K.K.); 2Department of Nanomedicine and Dermatology, California Innovation Corporation, San Diego, CA 92113, USA; menaateam@gmail.com; 3Faculty of Pharmacy, Vasile Goldiș Western University, 310045 Arad, Romania; neliolah@yahoo.com; 4Department of Pharmaceutics, College of Pharmacy, Jazan University, Jazan 45142, Saudi Arabia; scisp2@aliyun.com; 5Department of Pharmaceutical Sciences, College of Pharmacy, Princess Nourah Bint Abdulrahman University, Riyadh 11564, Saudi Arabia; bobdd363@aliyun.com; 6Department of Pharmaceutical Sciences, College of Pharmacy, Shaqra University, Shaqra 15273, Saudi Arabia; mxx238@aliyun.com

**Keywords:** *Linum*
*usitatissimum*, nutraceutical, leishmaniasis, water-in-oil cream

## Abstract

Leishmaniasis, remains a serious health problem in many developing countries with thousands of new cases recorded annually. Novel therapies are required as existing treatment regimens are limited by their high cost, high toxicity, increased parasite resistance, patient’s intolerance, and invasive means of long-duration administration. With several studies reporting the anti-leishmaniasis promise of medicinal plants, interest in plants and herbal drugs is attracting much attention worldwide. In this pilot study, we analysed extracts of *Linum usitatissimum* seeds (LU) to identify essential phytochemicals and test their activity against cutaneous leishmaniasis both in-vitro and in-vivo. We performed phytochemical screening of LU seeds extract as well as its in-vitro leishmanicidal and anti-amastigote assays. Water-in-oil cream containing 10% LU crude extract (10 mg/mL) was then prepared. The stability of the cream was evaluated for 28 days at 8 °C, 25 °C and 40 °C. In-vivo efficacy and safety of the cream was performed in 26 patients with cutaneous leishmaniasis who agreed to participate voluntarily in the study. The active treatment period lasted for 3 weeks, while the follow-up period was extended to 4 months. During the active study period, images of skin lesions were taken before and after treatment. Analyses of LU seeds extract confirmed the presence of terpenoids, flavonoids, tannins, alkaloids, and polyphenols. In-vitro studies showed significant activity against promastigote and intracellular amastigote forms of *Leishmania*
*major*. The cream was pharmaceutically stable, although some minor changes were noticed in relation to its physical characteristics. In-vivo assessment of the cream showed a 69.23% cure rate with no side effects, allergy, or irritation. We conclude that our newly developed water in oil cream containing 10% LU seeds extract could be an effective and safe topical anti-leishmanial medication for patients with CL.

## 1. Introduction

Leishmaniasis, a vector-borne disease categorized as a class I (emerging and uncontrolled) disease by the World Health Organization, is a serious and increasing threat caused by a protozoan belonging to the genus *Leishmania* that lives in the tissue and blood of the host [1,2,3,4]. This parasitic infection is transmitted through the bite of phlebotomine female sand flies. Leishmaniasis is primarily an infection of animals; however, it can affect humans if infected animals, humans and the vector occur in the same environment [1].

*Leishmania* parasites are divided into Old World species (Mediterranean basin, Middle East, Africa, India) which comprise *L. major*, *L. infantum*, and *L. tropica*, and New World species (Middle and South America) which include *L. amazonensis*, *L. chagasi*, *L. mexicana*, *L. naiffi*, *L. braziliensis*, *L. donovani* and *L. guyanensis* (Middle and South America) [2]. In addition to these species, all 31 are not mentioned here in the manuscript, including 20 human pathogenic species [5].

The severely neglected disease is endemic in more than 89 countries with an estimated 350 million people at risk, an overall prevalence of 12 million cases, and an annual underestimated incidence of about 2–2.5 million cases [1,2]. The three important clinical forms of leishmaniasis are visceral, cutaneous and mucocutaneous [1,2]. Cutaneous leishmaniasis and visceral leishmaniasis are known to be the two major forms of leishmaniasis in humans [2,3,5].

Cutaneous leishmaniasis is a chronic granulomatous infection of reticular endothelial cells of skin and represents the most prevalent form of leishmaniasis with 90% of all cases occurring in Pakistan, Algeria, Brazil, Peru, Afghanistan, Iran, Iraq, Syria, and Saudi Arabia [2,3]. The cutaneous leishmaniasis infection rate was estimated to be 4.6 cases per 1000 persons every year [3]. Cutaneous leishmaniasis, more popularly known as Oriental Sore or Delhi Boil in Pakistan, is widely distributed in all Pakistani provinces where it is estimated to affect 15,000–20,000 people annually [3,5]. Although it was assumed that *L. tropica* and *L. major* were the respective aetiological agents of dry and wet cutaneous leishmaniasis lesions, it was eventually reported that the manifestations of the skin lesions do not indicate the aetiology of species of *L**eishmania* [3]. Species of *Leishmania* vary in their sensitivity to the limited available therapies. The identification of the infecting *L**eishmania* parasite used to be laborious compared to many other infectious diseases, and the follow-up of patients and long-term therapeutic effects are often problematic [2,6]. As a possible consequence, most clinical treatment trials have been designed and reported poorly, resulting in a lack of financial incentive for pharmaceutical companies to invest in the development of drugs for a disease that is believed to primarily affect people that lack financial resources [2,7].

The currently available treatment for tegumentary leishmaniasis relies on pentavalent antimony as the first-line therapy, and pentamidine, amphotericin B or paromomycin as second-line treatments. However, it is worth noting that the drugs used for treating leishmaniasis present several side effects (e.g., high toxicity, increased resistance, intolerance) and limitations (e.g., high cost, complex therapeutic scheme, relatively low efficacy) [5,8]. There is dire need for new effective treatments with no or minimum side effects. Therefore, intensified research programs to rationally design innovative drugs, while improving vector control and leishmaniasis diagnostics, are needed [2,9].

*Linum usitatissimum* (LU), also known as flaxseed or linseed, is an annual herb with shallow root system that belongs to the genus *Linum* in the family Linaceae [10,11]. Interestingly, *L. usitatissimum* is a functional food that represents the richest natural grain source of lignans and accumulates substantial amounts of other health beneficial phytochemicals such as alkaloids, steroids, coumarins, saponins, tannins, terpenoids, glycosides (e.g., secoisolariciresinol diglucoside), linolenic acid, unique proteins, phenolic acids (e.g., hydroxycinnamic acids) and flavonoids (e.g., flavonols). For instance, oil from *L. usitatissimum* contains 40–50% of α-linolenic (ω-3) acid, and rich in phytosterols and tocopherols [12,13,14,15]. These bioactive components can improve the human immune system and prevent inflammation, with particular chemo-preventive actions toward cancer, diabetes mellitus and cardiovascular diseases, possibly thanks to their high antioxidant capacity [16,17,18,19]. Recently, it was suggested that LU-derived phytochemicals with powerful antioxidant properties, such as flavonols and hydroxycinnamic acids, might be of particular interest for dermatologic applications [20]. A study conducted by Abbasi et al. [21], fabricated Zinc oxide (ZnO) nanoparticles of flax by green approach. They concluded that ZnO nanoparticles exhibited potent antileishmanial activity against the parasite *L. major* [21]. Similarly, another study conducted by Al-Sugmiany et al. [22] detected the genetic and morphological effect of flaxseed extract on the *L. tropica*. They concluded that the chloroform extract of flaxseed has a high effect on the genome of parasite during the reproductive process that leads towards the mutation and resulted in high damage to the parasite’s genetics [22].

Topical drug delivery is localized delivery of drug moiety to the surface of skin for the treatment of various cutaneous infections. It has the advantage of painless delivery of drugs to the target site and ideally produces effective drug concentration. The side effects of several drugs can be reduced when they are delivered via topical or transdermal route [23].

The present study sought to determine the in-vitro and in-vivo anti-leishmanial effects of *L. usitatissimum*. For this purpose, 10% of LU crude extract was easily, rapidly, and cost-effectively encapsulated into a developed water in oil (W/O) cream, which was then physico-chemically characterized and tested in patients suffering from cutaneous leishmaniasis.

## 2. Results and Discussion

### 2.1. Phytochemical Screening: Qualitative and Quantitative Evaluations

The phytochemical analysis of extracts from seeds of *Linum usitatissimum* revealed the presence of important phytochemicals as shown in Table 1. These secondary-derived plant metabolites include phenols, terpenoids, flavonoids, tannins, and alkaloids. These results are in line with previous studies [21,22], although in our experimental conditions, we did not detect the presence of saponin, or glycosides, as previously reported [22]. Interestingly, alkaloids, terpenoids, steroids, lignans, and flavonoids (phenols) present in some plants exert an anti-leishmanial activity [20,21,22]. The total phenolic content (TPC) of LU seeds extract, which is directly related to the antioxidant activity, was milligram of gallic acid equivalent (mg of GAE/100 g), which represents a great amount in agreement with previous findings [21].

### 2.2. Stability of Cream

In the preparation of any pharmaceuticals, the physical properties are considered important and a cream having high consistency, elegancy and stability is preferred [24]. The LU extract-based cream was kept at different storage condition i.e., at 8 ± 0.1 °C (in refrigerator), 25 ± 0.1 °C (at room temperature) and 40 ± 0.1 °C (in incubator) for 28 days and examined physically from time to time for consistency, liquefaction, colour change, and cracking. The cream was yellowish in colour having a smooth elegance. There was no phase separation after centrifugation at 10,000 rpm. The pH is an important factor for topical formulations, and it must range between 5 and 6 to avoid skin irritation [24]. The results revealed that there were insignificant changes in the pH of cream kept at different temperatures for 28 days.

### 2.3. Viscosity of Prepared LU Cream

Viscosity has an important role in the delivery of drugs used via topical or transdermal application. Various parameters like stability, spreadability, drug release, and ease of application depend on the viscosity [24,25]. The viscosities of the LU cream were analysed at different times and temperatures and are presented in Table 2. There was a little variation in the viscosity of the LU cream kept at different temperatures, however it was insignificant i.e., *p* > 0.05.

### 2.4. In-Vitro Leishmanicidal Activity of LU against L. major Promastigotes

In-vitro anti-leishmanial activity of the LU crude powder, LU crude extract, and liquefied LU cream were tested against promastigotes of *L. major*. Amphotericin-B, a drug effective against pentavalent antimony-resistant mucocutaneous disease and visceral leishmaniasis, while albeit costly, toxic and poorly-tolerated in its original form [25,26,27,28], was used as a common reference drug [19]. Among the few studies assessing the anti-cutaneous leishmaniasis activity of plant extracts, most used *L. tropica* [20,21] or *L. major* [19,21], and in far fewer cases *L. viannia panamensis* [22]. In Pakistan, the approximately 21,000–35,000 reported cases were both anthroponotic and zoonotic forms of cutaneous leishmaniasis, which were attributed to *L. tropica* (frequent in northwest) and *L. major* (common in south), respectively [5]. As shown in Table 3, liquefied LU cream showed the anti-leishmanial activity with 18.23 µg/mL compared to LU crude extract with IC_50_ of 19.51 µg/mL (*p* < 0.05) and LU powder of IC_50_ of 22.93 µg/mL (*p* < 0.05). Interestingly, the elevated inhibitory activity of LU forms observed in this study against *L. major* parasites, after 24 h incubation, provides evidence and basis for their potential use as therapeutic agents against leishmaniasis. For instance, our data showed that the IC_50_ of LU seeds powder or LU seeds extract was better than the IC_50_ of water (279.48 µg/mL) and methanolic (42.82 µg/mL) extracts of leaves of *Aloe secundiflora*, which also significantly inhibited the growth of *L. major* parasites as compared to amphotericin B with respect to the parasite infection rates [19]. In addition to exert the greatest activity, which was dose dependent (data not shown), the newly developed cream formulation presented the advantages, over the crude and powder extracts, to be thermostable over 28 days (data not shown) and topically applicable.

### 2.5. In-Vitro Leishmanicidal Activity of LU Seeds Extract against Intracellular Amastigotes

As previous anti-leishmanial activity of LU seeds crude extract provides insignificant differences compared to that of the liquefied LU cream, we decided to pursue our in-vitro investigations with the LU crude extract. Thereby, after 48 h incubation, the leishmanicidal effect of LU seeds crude extract on amastigotes of *L. major* was evaluated in J774 macrophages. It was noted that IC_50_: 21.8 ± 4.5 μg/mL is only one third of the potency of reference drug amphotericin B, which was determined as 8.6 ± 1.9 µg/mL. Furthermore, this activity, which was dose dependent (data not shown), was promising when compared to other studies that assessed the in vitro cutaneous leishmanicidal effect of plant extracts. For instance, it was recently shown that low doses of macerated ethanolic extract of aerial parts of *Euphorbia petiolata* killed promastigotes of *L. major* dose-dependently with IC_50_ of 0.123 mg/mL after 48 h incubation [29,30].

### 2.6. In-Vivo Anti-Leishmanial Efficacy of LU Seeds Extract in a Clinical Setting

The in-vivo effect of LU cream was evaluated for three weeks in relatively young patients with an age range of 5–20 years with cutaneous leishmaniasis. Measurements and calculations for all lesions and indurations were recorded separately (Table 4). Briefly, the mean number of lesions per patient was 1.2 and mean lesion size on the patients was 25.3 ± 0.1 mm. The pre-treatment and post-treatment photographs of the patients (Figure 1) allowed evaluating the cure rate which was 69.23% (Table 4). Such promising clinical outcome corroborates the in-vitro anti-leishmanial efficacy of LU seeds extract and may be explained by the presence in LU seeds extract of alkaloids and peptides to greatly contribute to the anti-leishmanial activity [2]. Interestingly, no side-effects, such as allergy patch test reactions, were reported during the 3-months post-treatment. The treatment failure in 30.77% may be explained by several reasons including involvement of various species of leishmania, individual (pharmaco) genetic variations or secondary infections.

## 3. Materials and Methods

### 3.1. Reagents and Instruments

The reagents employed in this study included: ethanol (analytical grade) and n-hexane (Madina Chemicals Ltd., Lahore, Pakistan); ABIL EM-90^®^ (Evonik, Darmstadt, Germany); liquid paraffin oil and bees wax (Merck, Darmstadt, Germany); Benedict’s reagent and Wagner’s reagent (Faculty of Pharmacy, Gomal University, Dera Ismail Khan, Pakistan); dimethyl sulfoxide (DMSO), chloroform, ether and methanol (Merck KGaA, Darmstadt, Germany); hydrochloric acid, sulphuric acid, ammonia, sodium carbonate, gallic acid, amphotericin B, folin and Ciocalteu’s reagent, a MTT (3-(4,5-dimethylthiazol-2-yl)-2,5-diphenyltetrazolium bromide) assay kit (Sigma-Aldrich, New York, NY, USA); ferric chloride and aluminium chloride (Gebinde, Hamburg, Germany); NNN growth medium, foetal calf serum (FCS) and RPMI 1640 (Dutscher, Paris, France).

The following instruments were used in this study: a grinder POLYMIX^®^ PX-MFC 90D (VWR™/Avantor, Radnor, PA, USA); a centrifuge (Kukusan H-200, Okazaki, Japan); a cold incubator (Shellab 2020-2E, New York, NY, USA); pH/mV/temperature/ion/conductivity meter Denver Instrument 9357.1 (Cole Parmer, Paris, France); UV/VIS Spectrophotometer V-630 (Jasco GmbH, Berlin, Germany); a digital humidity meter (TES Electrical Electronic Corp., Taiwan); an electrical balance ELB300 (Shimadzu Corp. Tokyo, Japan); a mixer/homogenizer/shaker (Waring Commercial, Birmingham, UK); an incubator IF55 and oven UF30plus (Memmert GmbH, Darmstadt, Germany); a SpectraMax Paradigm Multi-Mode Detection Platform (Molecular devices LLC, Washington, USA); a refrigerator (PEL, Karachi, Pakistan); a water bath (Stuart RE300DB) and rotary evaporator (Stuart RE300) (Keison Products, London, UK); pH meter (Systronic, Mumbai, India) and rheometer (NDJ, K8, Seoul, Korea).

### 3.2. Plant Collection and Preparation of Crude Plant Extracts

Seeds of *Linum usitatissimum* were purchased from a local market of Dera Ismail Khan, Pakistan, and authenticated by the taxonomist Dr. Mushtaq (Quaid. I. Azam University, Islamabad, Pakistan). A specimen was deposited in the herbarium under the voucher number 17/012017/E.

About 250 g of LU seeds were crushed and the resulting powder was then macerated into 1 l of 80% ethanol for 48 h. Aluminium foil was used to cover the beaker, which was subsequently kept in the laboratory at room temperature. Shaking of the macerate was done for 10 min after every 12–24 h [22], followed by coarse filtration using several layers of muslin cloth and a fine filtration carried out through a Whatman # 01 filter paper. The resulting filtrate was evaporated, in a rotary vacuum evaporator set up at 40 °C, until the volume of the concentrate was reduced to one third of the original volume, and immediately transferred to an oven for drying completion.

The crude extract in powdered form was eventually stored at 8 °C in an airtight container. The percent yield of the crude extract, calculated based on the weight before (i.e., 250 g) and after extraction (i.e., ~100 g), was 40% on average. Dilutions of either LU powder or crude extract were made in DMSO (solvent) whenever required.

### 3.3. Phytochemical Screening

The phytochemical screening was carried out using standard procedures as previously described by Safowora (1993) and Trease and Evans (2002) [27,28].

#### 3.3.1. Total Phenolic Content

The total phenolic content (TPC) was estimated by the method reported by Anwar and Przybylski (2012) with some modifications [29,30]. Briefly, 0.5 mL of the diluted crude extract (2 mg/mL) was mixed with 2.5 mL of Folin–Ciocalteu reagent (FCR) and 2 mL of Na_2_CO_3_ (75 g/L). The incubation of the mixture was done at 50 °C for 5 min, and then cooled to room temperature. Intense blue colour was developed. The absorbance was then measured by spectrophotometry at 760 nm. Gallic acid (0, 10, 20, 40, 60, 80, 100 µg/mL) was used as a standard. The results, obtained from three independent experiments, were expressed as mg of gallic acid equivalent weight (mg) per 100 g of dry mass (mg GAE/100 g DM).

#### 3.3.2. Terpenoids

For the presence of terpenoids, 2 mL of crude extract of LU was treated with an equal amount of chloroform and concentrated H_2_SO_4_.

#### 3.3.3. Flavonoids

The presence of flavonoids was confirmed by dissolving 2 mL of LU extract in an equal amount of 10% lead acetate.

#### 3.3.4. Tannins

For the presence of tannins, about 2 mL of crude extract of LU was mixed with 0.1% ferric chloride solution.

#### 3.3.5. Saponins

The presence or absence of saponins in the LU extract was confirmed by treating 2 mL of the crude extract of LU with an equal volume of Benedict’s reagent.

#### 3.3.6. Glycosides

The crude extract of LU was hydrolysed with HCL solution and then neutralized by NaOH solution. A few drops of Fehling’s solution were also added to confirm the presence of glycosides.

#### 3.3.7. Alkaloids

About 2 mL of the LU extract was added to an equal amount of Wagner’s reagent. A brownish precipitate confirmed the presence of alkaloids in the extract.

#### 3.3.8. Reducing Sugar

To confirm the presence of reducing sugars in the LU extract, the crude extract of LU was mixed with distilled water and filtered. Fehling’s solution and filtrate was boiled for a few minutes.

### 3.4. Preparation of LU-Based Emulsion for Topical Application

The W/O cream was prepared by adding small water droplets dispersed in a continuous oily phase, under constant agitation [30,31]. The oily phase, which comprised paraffin oil, bees wax and surfactant ABIL-EM 90, was heated up to 75 ± 5 °C. In parallel, the aqueous phase that consisted of distilled water (D/W) was heated in a water bath at the same temperature. The LU seeds extract (10%, i.e., 10 g) was then added into the aqueous phase. The aqueous phase was then added to the oily phase by dropping. Stirring was done at 2000 rpm for 15 min using a mechanical mixer. The mixer speed was reduced to 1000 rpm for 5 min after the complete addition of the aqueous phase. For a nice fragrance, a few drops of aqua blue were added (to the best of our knowledge, no reports have been published reporting anti-leishmania activity of aqua blue per se). Eventually, the speed of the mixer was further reduced to 500 rpm for 5 min for complete homogenization, until the emulsion cooled to room temperature. The final composition is listed in Table 5.

### 3.5. Pharmaceutical Stability Test

Stability tests of the LU powder, the LU crude extract, and the liquefied LU cream were performed at different storage temperatures. These tests were performed on samples kept at 8 ± 0.1 °C (in refrigerator), 25 ± 0.1 °C (at room temperature) and 40 ± 0.1 °C (in incubator) at various time intervals over 4 weeks (i.e., 12 h, 24 h, 36 h, 48 h, 72 h, 7 days, 14 days, 24 days, and 28 days) [26]. The stability of the cream, including a heat and cool cycle, a freeze thaw cycle, and centrifugation, was also evaluated as per ICH (International Conference on Harmonization) guidelines. Moreover, the pH of the LU extract loaded cream was also evaluated using a calibrated pH meter (Systronic, Kunshan, China) for period of 28 days.

### 3.6. Viscosity Measurement of Cream

The viscosity of the prepared formulation (cream) was determined by using viscometer (NDJ, K8, Seoul, Korea) at 25 °C. The readings were taken at specified times in triplicates and results were averaged.

### 3.7. In-Vitro Leishmanicidal Activity of LU against Promastigotes

The effect of *Linum usitatissimum* (LU powder, LU crude extract, or LU cream liquefied by dilution in DMSO) was evaluated in logarithmic phase promastigotes (5 × 10^6^ promastigotes/mL) of *L. major* using a microdilution method in a 96-well microtiter plate [32]. A volume of 5 × 10^5^ promastigotes (100 µL) was seeded per well. About 80 µL of NNN growth medium supplemented with FCS (20%) and RPMI 1640 medium were added in all wells. With the exception of the full first line of wells, 20 µL (20 µg) of the crude extract (stock concentration: 1 g/L) was subsequently added and properly mixed before 7 two-fold serial dilutions were made using an aliquot of 100 µL from each previous “test” well. Thus, the final volume was maintained to 200 µL. The concentration of the LU bioactive ingredient was equivalent between the LU crude powder, LU crude extract, or LU liquefied cream, and in the first “test well” such concentration was 100 µg/mL and the last well contained 0.78 µg/mL. The last two empty wells remaining from the “test” lines were used for negative and positive controls, which contain DMSO solution or the same quantity of the reference drug amphotericin B (stock concentration solution: 0.2 mg/mL), respectively. After incubation at 25 °C in 5% CO_2_ for 24 h, the number of (treated or untreated) promastigotes was counted using a hemocytometer through a light microscope, by placing a drop of the respective culture on the slide. The percent (%) inhibition obtained with the test sample was compared to that of controls using the following formula:100 − (Number of parasites (test sample)/Number of parasite (control sample)) × 100(1)

IC_50_ values, i.e., the concentration of the crude extracts required to inhibit 50% of promastigote’s growth, obtained by sigmoidal dose-response curve analysis using the scatter plot option of Graph Pad Prism 5.0 software, were expressed as the mean of samples ± standard deviation (STD) from three independent experiments conducted in duplicates.

### 3.8. In-Vitro Leishmanicidal Activity of LU Seeds Extract against Intracellular Amastigotes

The leishmanicidal effect of LU seeds crude extract on *L. major* amastigotes was evaluated in J774 macrophages. First, 106 cells/mL of this commercial macrophage strain were seeded in 24-well plates containing RPMI medium supplemented with 10% (heat-inactivated) FBS, 15 μg/mL of gentamicin, and 1000 u/mL of penicillin). The plates were then incubated at 37 °C in 5% CO_2_ for 2 h, and the non-adherent cells were removed by 1× phosphate-buffered saline (PBS) washing. Next, stationary-phase *L. major* promastigotes were added to the macrophages at a ratio of 40:1 (parasite: macrophage) and incubated for 4 h at 37 °C in 5% CO_2_. During this period, the parasites invaded the macrophages and then transformed into amastigotes. Free promastigotes were subsequently removed by successive washes with 1× PBS. Infected macrophages were then treated with LU seeds extract (pre-resuspended in DMSO), at concentrations ranging from 100 μg/mL and following 4 two-fold dilutions carried out using 1 mL each time to reach 6.25 μg/mL, before incubation for 48 h. Amphotericin B (stock concentration solution: 0.2 mg/mL), was used as a reference drug control. Infected treated macrophages were then washed with 1× PBS, fixed with absolute methanol for 10 min, and stained with 5% Giemsa in PBS for 25 min before their observation under a light microscope. Positive staining was directly proportional to the concentration of surviving parasites. Anti-leishmanial activity was evaluated by observing 250 macrophages within each treatment group. The percentage of infected macrophages was calculated using the following formula:(Number of infected macrophages/250 macrophages observed) × 100(2)

Infection index values were then calculated according to the following formula:Percentage of infected macrophages × average number of intracellular amastigotes per infected macrophage(3)

Infection index values were then converted to percentage survival values relative to the untreated parasite population. The IC_50_ values, obtained by sigmoidal dose-response curve analysis using the scatter plot option of Graph Pad Prism 5.0 software, were expressed as the mean of samples ± standard deviation (STD) from three independent experiments conducted in duplicates.

### 3.9. In-Vivo Anti-Leishmanial Efficacy of LU Seeds Extract

The in-vivo effect of the LU cream (undiluted W/O emulsion) was evaluated in Pakistani male and female individuals mostly (66%) with zoonotic form of cutaneous leishmaniasis. There were 26 volunteers having an age range of 5–20 years and mainly (82%) originated from southern part of Pakistan. Prior to the start the clinical trial and applying the product application, a dermatologist examined the person to confirm leishmaniasis. Volunteers were not informed about the content of the cream. The inclusion criteria for volunteers in the study were: confirmed case of cutaneous leishmaniasis, absence of any product used during the treatment for the same purpose. Also, all volunteers were asked to read and sign the recruitment protocol and asked to report every week for a period of 12 weeks from the beginning of the study.

On the first day, a patch test (Burchard test) was performed on their forearms to determine any possible unwanted/undesirable effects/sensitivity (e.g., allergy patch test reactions) to the cream formulation. The volunteers were instructed to apply the cream for 3 weeks. Every individual was instructed to come every week, during the study period, for the skin measurements/lesion(s) observation. After sensitivity test, measurements and pictures were taken during every visit. 

### 3.10. Ethical Consideration

The trial was conducted in accordance with the Declaration of Helsinki (1964), and the protocol was approved by the Board of Advanced Studies and Research (BASR) GU D.I. Khan. (Reference No. 164-165/Acad/GU). All the participants were well informed about the trial and a written consent was obtained from their parents

### 3.11. Statistical Analysis

The SPSS version 21 software was used for the processing of experimental data. Mean of samples ± standard deviation (STD) from at least three independent experiments were performed. Student’s *t*-test was used for comparing cream and standard. *p* < 0.05 was considered statistically significant.

## 4. Conclusions

It can be concluded from the findings of this study that seed extract of *Linum usitatissimum* has promising anti-leishmanial properties in-vitro and in-vivo. This activity may be attributed to presence of different phytochemicals that are reported to possess anti-leishmanial activity. It was also possible to develop a novel topical formulation (cream) comprising 10% hydro-alcoholic extracts of seed crude extract of *Linum usitatissimum* that had no side effects, unlike those available treatments on the market. Finally, the findings offer a scientific basis to promote the use of natural products for the treatment of chronic cutaneous leishmaniasis.

## Figures and Tables

**Figure 1 antibiotics-11-00432-f001:**
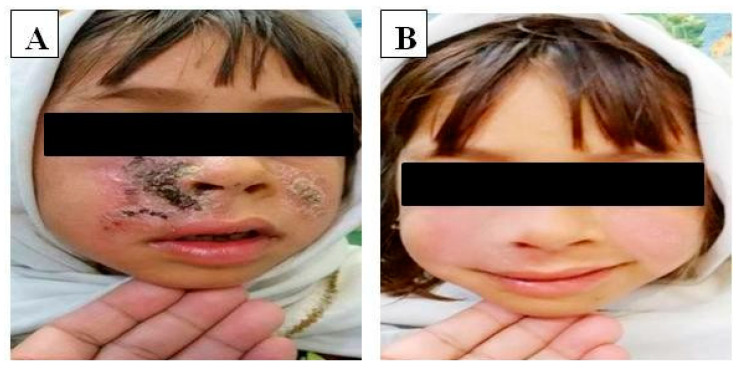
Effects of LU cream on patients (**A**; Before, **B**; after) with cutaneous Leishmaniasis.

**Table 1 antibiotics-11-00432-t001:** Phytochemicals screening in seeds extract of *Linum usitatissimum*.

Phytochemicals	(*Linum usitatissimum*)
Phenols	+
Terpenoids	+
Flavonoids	+
Tannins	+
Saponins	−
Glycosides	−
Alkaloids	+
Carbohydrates	−

Present: +, Absent: −.

**Table 2 antibiotics-11-00432-t002:** Viscosities (Centipoise) of the LU cream at the indicated temperature and time.

Time	Viscosities at 8 °C	Viscosities at 25 °C	Viscosities at 40 °C
**Day 0**	13,380	13,380	13,380
**Day 1**	13,380 ± 13.8	13,370 ± 12.1	13,201 ± 10.2
**Day 2**	13,378 ± 12.6	13,340 ± 12.3	13,100 ± 11.3
**Day 7**	13,350 ± 13.3	13,310 ± 12.7	12,900 ± 11.3
**Day 14**	13,342 ± 13.4	13,290 ± 11.6	12,870 ± 11.5
**Day 28**	13,337 ± 12.2	13,270 ± 11.8	12,700 ± 13.2

**Table 3 antibiotics-11-00432-t003:** Anti-leishmanial activity of LU seeds powder, crude extract, and liquefied cream against *L. major* promastigotes using amphotericin-B.

Test Samples	IC_50_ (µg/mL ± STD)	Maximum LU. Inhibition (%)	Amphotericin-B Inhibition (%)	Negative Control Inhibition (%)	Comments
Crude powder	22.93 ± 0.2	67.85	100	0	LAA
Crude extract	19.51 ± 0.3	69.91	100	0	IAA
Liquefied Cream *	18.23 ± 1.1	73.85	100	0	HAA

LAA: lowest antileishmanial activity, IAA: Intermediary antileishmanial activity, HAA: Highest antileishmanial activity, STD: standard deviation. * The maximum LU inhibition of liquefied cream was at a dose of 100 µg/mL.

**Table 4 antibiotics-11-00432-t004:** In-vivo study on *Linum usitatissimum* cream on young patients with cutaneous leishmaniasis.

Number (#) of patients	26
Mean # of lesions per patient	1.2
Mean lesion size SD (mm)	25.3 ± 0.1
# of cured patients	18
# of patients for whom therapy failed	8
Cure rate in percentage (%)	69.23%

3/8 of patients showed a relative improvement but a disease relapse was not discarded; 5/8 of the patients failed completely to the treatment.

**Table 5 antibiotics-11-00432-t005:** Composition of cream (% *w*/*w*).

	Ingredients	Percent Used
**01**	^α^ Liquid Paraffin	16
**02**	^α^ ABIL EM-90^®^	5
**03**	^α^ Bees wax	4
**04**	^β^ Plant Extract	10
**05**	^β^ Fragrance	1
**06**	^β^ D/W	64

^α^; Oil Phase, ^β^; Aqueous Phase.

## Data Availability

In-vitro and in-vivo research data that were used to evaluate a novel W/O botanical cream support the findings of this study and are included within the article with free access.

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
