# Peer review of "Fabrication and Evaluation of W/O Emulsion Loaded with Linum usitatissimum Seeds Extract for Anti-Leishmaniasis Efficacy"

_antibiotics, 2022, doi:10.3390/antibiotics11040432_

Round 1

Reviewer 1 Report

This is a fascinating study.  The authors have provided some evidence that suggests that an easily prepared extract of a common plant may be effective against a disease affecting many people, one that is disfiguring.   The technology is relatively unsophisticated,  but better was likely not available. And the results are significant and could lead to better and cheaper treatment by further development.

  However, I have several questions relevant to the science:

A. The protocol for the patients asked them to go home and apply the cream, presumably daily; were there any checks to assure that they did so? If application were not consistent, the results could have been skewed, most likely against the positive results observed.

B. It appears as though there were no blinds in the in vivo test.  Were any patients given cream that contained none of the extract?

C.  All of the subjects were under 20 and many were under 16.  How was permission given to test the children?  Were parents involved?

D. Why were there no subjects over 20 in the in vivo testing? Surely there were individuals with the disease.

I think it advisable to address these questions in a revised manuscript.

Here are a few editorial comments and suggestions:

  1. l.44  Change "the" to "an"
  2. l. 51  Change "Not only these species are known but almost" to "In addition to these species"
  3. l. 52 Change "human's" to "human"
  4. l. 80 Use "drugs"
  5. l. 84 "accumulates"
  6. l. 88 Change "which to "and"
  7. l. 102 Delete "mainly"
  8. l. 135  Shouldn't it be p<.05?
  9. l. 218 "powdered extract (namely, crude extract)" is confusing and cumbersome.  Suggest "...crude extract in powdered form"
  10. l. 282-284  I think this is just a missed carriage return.

Author Response

Query A. The protocol for the patients asked them to go home and apply the cream, presumably daily; were there any checks to assure that they did so? If application were not consistent, the results could have been skewed, most likely against the positive results observed.

Reply: The patients enrolled in the study were properly followed up for the application of cream. They were contacted telephonically on daily basis for checks and to assure that they did so.

Query B. It appears as though there were no blinds in the in vivo test.  Were any patients given cream that contained none of the extract?

Reply: There was no single or double blind testing in this study. All the patients were treated with LU extract loaded cream. We can’t treat any patients with blank formulation (without LU extract or any other active drug) because it can be risky and the infection may propagate.

Query C.  All of the subjects were under 20 and many were under 16.  How was permission given to test the children?  Were parents involved?

Reply: Before starting of treatment, permission was granted from their parents and written consent was also obtained. This statement has been added in the revised manuscript.

Query D. Why were there no subjects over 20 in the in vivo testing? Surely there were individuals with the disease.

Reply: We selected all the available infected patients during the study period and we described the actual age range.

Here are a few editorial comments and suggestions:

  1. l.44  Change "the" to "an"

 (Changed as instructed)

  1. l. 51  Change "Not only these species are known but almost" to "In addition to these species" (Changed as instructed in the revised MS)
  2. l. 52 Change "human's" to "human"

(Changed)

  1. l. 80 Use "drugs"

(Modified)

  1. l. 84 "accumulates"

(Modified)

  1. l. 88 Change "which to "and"

(Changed)

  1. l. 102 Delete "mainly"

(Deleted)

  1. l. 135 Shouldn't it be p<.05?

It was a typo error that has been rectified in the revised MS.

  1. l. 218 "powdered extract (namely, crude extract)" is confusing and cumbersome.  Suggest "...crude extract in powdered form"

Changed as suggested by the reviewer

  1. l. 282-284  I think this is just a missed carriage return.

Modified in the revised MS

Reviewer 2 Report

The manuscript entitled “Fabrication and evaluation of W/O emulsion loaded with Linum usitatissimum seeds extract for anti-Leishmaniasis efficacy” regards the formulation of a

 water-in-oil cream containing 10%  Linum usitatissimum  crude extract  and its application in the treatment of cutaneous leishmaniasis.

Results obtained are interesting but the manuscript lacks some important information. The phytochemical analysis is based only on a qualitative screening of the class of compounds present in the extract. Moreover, methods used for this screening should be given in detail in the experimental part while only a reference to a general book is reported. Since these preliminary analyses revealed the presence of terpenoids, alkaloids and phenols, which could be responsible for the efficacy of the treatment, the principal compounds of these classes present in the extract should be identified with more specific chemical analysis such as for example HPLC and/or HPLC-MS.

Regarding in vivo tests, since 8 patients out of 26 failed the treatment it should be interesting to understand if there is a relation with the species of Leishmania parasites since volunteers were confirmed for leishmaniasis but nothing is said about the species.

Author Response

The manuscript entitled “Fabrication and evaluation of W/O emulsion loaded with Linum usitatissimum seeds extract for anti-Leishmaniasis efficacy” regards the formulation of a water-in-oil cream containing 10% Linum usitatissimum crude extract  and its application in the treatment of cutaneous leishmaniasis. Results obtained are interesting but the manuscript lacks some important information.

1- The phytochemical analysis is based only on a qualitative screening of the class of compounds present in the extract. Moreover, methods used for this screening should be given in detail in the experimental part while only a reference to a general book is reported.

Reply: According to the reviewer instruction, the phytochemical screening has been now explained in the revised manuscript.

2- Since these preliminary analyses revealed the presence of terpenoids, alkaloids and phenols, which could be responsible for the efficacy of the treatment, the principal compounds of these classes present in the extract should be identified with more specific chemical analysis such as for example HPLC and/or HPLC-MS.

Reply: this study is based with intention to explore LU in crude form for cutaneous Leishmaniasis in pharmaceutical dosage form. In future, we planned to isolate and purify the active constituents and formulate the dosage form from the purified compound to trace the compound responsible for this activity.

3- Regarding in vivo tests, since 8 patients out of 26 failed the treatment it should be interesting to understand if there is a relation with the species of Leishmania parasites since volunteers were confirmed for leishmaniasis but nothing is said about the species.

Reply: Reason for failure to the treatment in 8 patients was not analyzed or traced out.

Reviewer 3 Report

In this manuscript, the authors have analyzed the extracts of Linum usitatissimum seeds (LU) to identify essential phytochemicals and tested their activity against cutaneous leishmaniasis (CL) both in- vitro and in-vivo. The work is interesting and can be accepted for publication after the author should consider the following comments for improvement.

  1. The manuscript requires comparison with available methods to treat leishmaniasis (table)
  2. The authors provide information about the presence of groups of chemical substances in the extract, however, the exact composition of the extract up to the metabolites name was not performed. It can identify the potential active ingredients and can be helpful to prevent long term possible side effects (discoloration, bruising).
  3. Cream pH measurements are also required (with time) to assess the stability of chemicals in the cream.
  4. Additional information is required on the physical cream properties such as droplet size, microscopy (confirming droplet size and distribution), rheological behavior (viscosity).
  5. Please provide a brief comparison of different drug delivery systems in the introduction part, so that readers could have a better understanding. In this regard, the following reference doi.org/10.1049/iet-nbt.2016.0106 could be helpful.
  6. It is important to study transdermal delivery of the prepared cream to assess the efficiency of the active substance delivery and to compare it with in vitro studies.
  7. Could the author please verify that they have provided the proper patient-consent documentation to the editors at the time of submission?

Author Response

  1. The manuscript requires comparison with available methods to treat leishmaniasis (table)

Reply: The only available therapy for the treatment of CL Leishmaniasis is subcutaneous injections of pentavalent antimoniate (Glucantime) which has already been explained in the manuscript

  1. The authors provide information about the presence of groups of chemical substances in the extract, however, the exact composition of the extract up to the metabolites name was not performed. It can identify the potential active ingredients and can be helpful to prevent long term possible side effects (discoloration, bruising).

Reply: The intention of the study was to evaluate the antileishmanial activity of crude extract of LU in pharmaceutical dosage form rather than its purification and isolation. The phytochemical screening was also performed however we didn’t perform the purification of metabolites.

  1. Cream pH measurements are also required (with time) to assess the stability of chemicals in the cream.

Reply: pH was determined in this study from time to time. The pH of the cream was 4.5-5.5 which is in the range of human skin pH. It has been now presented in the revised manuscript.

  1. Additional information is required on the physical cream properties such as droplet size, microscopy (confirming droplet size and distribution), rheological behavior (viscosity).

Reply: As we formulated conventional cream rather than nano-formulation, therefore we didn’t perform the Zetasizer analysis. However we had the viscosity data, which has been provided in the revised MS.

  1. Please provide a brief comparison of different drug delivery systems in the introduction part, so that readers could have a better understanding. In this regard, the following reference doi.org/10.1049/iet-nbt.2016.0106 could be helpful.

Reply: A brief introduction of topical drug delivery system has been provided in the revised manuscript. The paper suggested by the author has also been cited in the revised MS.

  1. It is important to study transdermal delivery of the prepared cream to assess the efficiency of the active substance delivery and to compare it with in vitro studies.

Reply: As the focus of the study was to treat the cutaneous Leishmaniasis (CL) rather than the visceral Leishmaniasis and CL can be treated topically therefore we didn’t perform any transdermal study.

  1. Could the author please verify that they have provided the proper patient-consent documentation to the editors at the time of submission?

Reply: Before start of the therapy, permission was granted from the parents of the patients. Furthermore written consent was also obtained.

Round 2

Reviewer 1 Report

Corrections and additions are noted and clear, as are explanations provided by the authors.

Author Response

Thank you very much for your valuable suggestions and queries which improved our manuscript a lot. 

Reviewer 2 Report

The authors only performed qualitatively analyses of compounds present in the extract with very old methods. The analytical methods available today are able to identify the main components of a crude extract which, in my opinion, is necessary to correlate the activity found. It is not possible to pospone this important part in a second manuscript.

Author Response

Thank you for sharing time to review our manuscript. It is worthy to mention here that all the qualitative methods were used for identification of major consituentis in LU extract because of availability of resources at our Lab. These methods are well established and considered as gold standards since long in literature.
It is also important to mention here that project has been closed for further analysis so new data generation is not possible.
The project was intended on crude extract rather isolated purified compounds and it has been compiled as per intension or objective of the project.

Moreover the introduction, results and conclusion have been improved in the revised MS. 

Reviewer 3 Report

As the author(s) clarified to the editors that they have obtained proper written permission from the parents of the patients, So, I conclude that this paper can be accepted for publication.

Author Response

Thank you for sparing time to review this manuscript and a good piece of suggestion was presented which improved our manuscript.